# Multiplexing cell-cell communication

John T Sexton[1] (ID) & Jeffrey J Tabor[1,2,*] (ID)

## Abstract

The engineering of advanced multicellular behaviors, such as the programmed growth of biofilms or tissues, requires cells to communicate multiple aspects of physiological information. Unfortunately, few cell-cell communication systems have been developed for synthetic biology. Here, we engineer a genetically encoded channel selector device that enables a single communication system to transmit two separate intercellular conversations. Our design comprises multiplexer and demultiplexer sub-circuits constructed from a total of 12 CRISPRi-based transcriptional logic gates, an acyl homoserine lactone-based communication module, and three inducible promoters that enable small molecule control over the conversations. Experimentally parameterized mathematical models of the sub-components predict the steady state and dynamical performance of the full system. Multiplexed cell-cell communication has applications in synthetic development, metabolic engineering, and other areas requiring the coordination of multiple pathways among a community of cells.

**Keywords** cell-cell communication; CRISPRi; genetic circuit design; multiplexers; synthetic biology

**Subject Category** Biotechnology & Synthetic Biology

**Mol Syst Biol. (2020) 16: e9618**

## Introduction

Synthetic biologists have long aimed to engineer cells to cooperate to perform complex tasks. For example, cell communities have been programmed to undergo synchronized gene expression dynamics (Brenner *et al*, 2007; Weber *et al*, 2007; Balagaddé *et al*, 2008; Danino *et al*, 2010; Chen *et al*, 2015), act as distributed computers (Tabor *et al*, 2009; Regot *et al*, 2011; Tamsir *et al*, 2011; Shin *et al*, 2020), and differentiate into simple patterns (Basu *et al*, 2005; Liu *et al*, 2011; Cao *et al*, 2016; Karig *et al*, 2018; Toda *et al*, 2018). However, due to challenges such as molecular cross-talk, no more than two communication systems have been combined in a single design (Scott & Hasty, 2016; Billerbeck *et al*, 2018; Kylilis *et al*, 2018). In stark contrast, evolution utilizes dozens of communication signals to orchestrate complex behaviors such as embryo patterning (Perrimon *et al*, 2012), the precise wiring of the nervous (Dickson,

2002), circulatory and lymphatic systems (Adams & Alitalo, 2007), and innate and adaptive immunity.

In electronic engineering, channel selectors (CSs) enable a single communication resource such as a wire to transmit multiple conversations. In a CS, a multiplexer circuit (MUX) is linked to a demultiplexer circuit (DEMUX), and both are controlled by a common SELECT signal (Fig 1A and B). The MUX receives $n$ input signals, but propagates only one to the DEMUX, depending on the SELECT value. The DEMUX relays this signal to one of $n$ possible outputs, again depending on SELECT. A conversation occurs because a given input ($IN_i$) is routed exclusively to the corresponding output ($OUT_i$). Multiple conversations can occur sequentially if the SELECT value is changed (Fig 1A and B).

Here, we demonstrate that CSs can be used to expand the communication capacity of biological systems. We begin by engineering a library of non-cross-reacting (orthogonal) transcriptional NOT and NOR gates based upon promoters with synthetic operator sequences and complementary single guide RNAs, enabling strong repression of those promoters via nuclease-dead Cas9. Next, we assemble those gates into 2-input MUX and 2-output DEMUX circuits in separate strains of bacteria. We then use two small molecule inducers to control the two MUX input signals, a third to select which of those two inducer signals is propagated through the MUX (i.e., act as a channel select line), and express an acyl homoserine (AHL) lactone-based cell-cell communication system as the MUX output. We go on to add an AHL-sensing module to the DEMUX and use the same channel select line to control which of its outputs is activated in response to AHL. Finally, we combine these two systems in strains grown in co-culture and demonstrate dynamical switching between two separate small molecule-mediated conversations. We combine experimental characterization with mathematical modeling to predict the function of each of the more complex systems from the sub-components that comprise it.

## Results

### Design of CS logic circuits

To multiplex cell-cell communication (Fig 1C and D), we implemented the simplest CS, where $n = 2$, genetically. First, we used formal logic synthesis to design the required 2-input MUX and 2-output DEMUX from the smallest possible number of Boolean NOT and NOR gates (Appendix Fig S1A and B). We restricted our design

1 Department of Bioengineering, Rice University, Houston, TX, USA
2 Department of BioSciences, Rice University, Houston, TX, USA
  *Corresponding author. Tel: +1 713 348 8316; E-mail: jeff.tabor@rice.edu

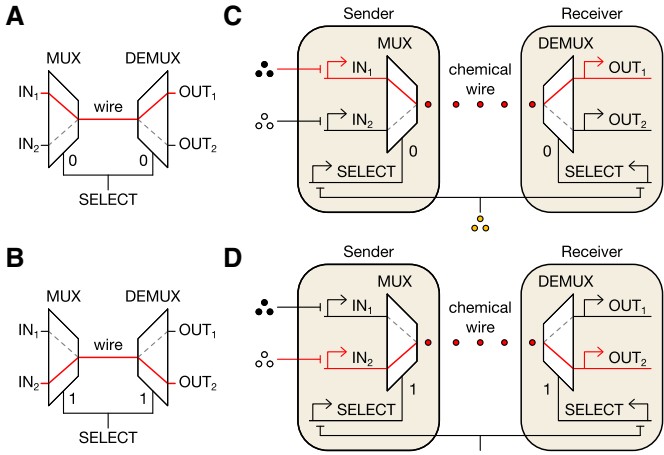

**Figure 1.  Electronic and biological channel selectors.**

A, B   In an electrical CS, the SELECT signal directs the MUX to choose an input to transmit across the wire and the DEMUX to route the signal to the corresponding output. The CS can thus transmit two separate conversations over the same wire by changing the value of SELECT.

C, D   Design of a genetically encoded CS that enables transmission of multiple conversations via a single cell-cell communication system (red circles). Here, $IN_1$, $IN_2$, and SELECT are transcriptional signals controlled by chemical inducers (black, white, and orange circles).

to NOT and NOR because these gates can be combined to achieve any digital logic operation and readily constructed in live cells using transcriptional repressors and repressible promoters (Tamsir *et al*, 2011; Stanton *et al*, 2014; Nielsen *et al*, 2016; Gander *et al*, 2017). Our design process yielded minimized MUX and DEMUX circuits composed of one NOT and three NOR gates assembled in three layers (Appendix Fig S1C and D), and two NOT and two NOR gates assembled in two layers (Appendix Fig S1E and F), respectively.

**Design and characterization of CRISPRi-based NOT and NOR gates**

To create the logic gates, we turned to the CRISPR interference (CRISPRi) technology, wherein a nuclease-dead Cas9 (dCas9):small guide RNA (sgRNA) complex sequence-specifically binds and represses transcription from a target promoter (Qi *et al*, 2013). We designed nine putatively orthogonal sgRNA:promoter pairs by encoding randomized and divergent operator sequences lacking homology to the *Escherichia coli* genome (Materials and Methods) between the −35 and −10 sites of otherwise constitutive promoters (Fig 2A and B). Indeed, all sgRNAs in our set (S1–S9) strongly repress their cognate promoters (P1–P9), without cross-repressing any of the eight non-cognate promoters (Fig 2C).

Each $Si$:$Pi$ pair constitutes a transcriptional NOT gate (NOT$i$) and inverts a high or low transcriptional input signal ($IN_{NOTi}$) into a low or high transcriptional output signal ($OUT_{NOTi}$), respectively (Fig 2D and E). To characterize each sub-component in our system, we designed a set of standard probe plasmids, wherein a promoter of interest drives transcription of an insulated superfolder green fluorescent protein (*sfgfp*) reporter gene (Appendix Fig S2 and Appendix Text S1). To generate a wide range of $IN_{NOTi}$ signals, we placed transcription of $Si$ under control of an anhydrotetracycline

(aTc)-inducible promoter system (i.e., an aTc sensor; Fig 2D) in the context of NOT$i$. Then, we co-transformed each aTc-NOT$i$ gate into *E. coli* pairwise with the aTc sensor probe plasmid and the corresponding P$i$ probe plasmid. We then exposed these 18 strains to different aTc concentrations and quantified the resulting $IN_{NOTi}$ and $OUT_{NOTi}$ signals via sfGFP fluorescence in calibrated molecules of equivalent fluorescein (MEFL) units (Appendix Fig S2). Most of the resulting $IN_{NOTi}$/$OUT_{NOTi}$ relationships (i.e., NOT$i$ transfer functions) are well fit by a constrained Hill model ($n = 1$), consistent with a recent study (Zhang & Voigt, 2018; Appendix Fig S3). However, an unconstrained Hill model better describes several of the transfer functions (Appendix Fig S3). Thus, we chose to use unconstrained Hill models to describe the behaviors of all NOT$i$ gates (Fig 2E, Appendix Table S1, Appendix Text S2). The root

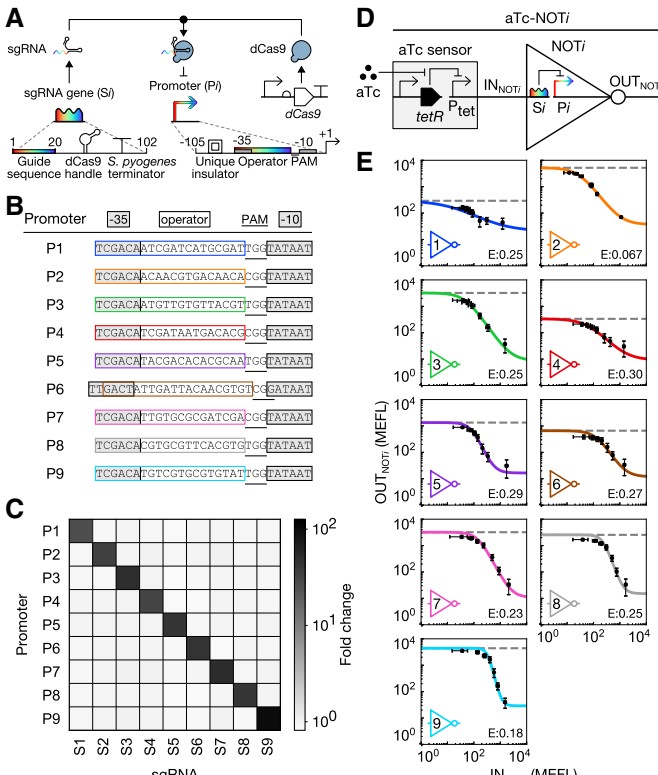

**Figure 2.  Design of a CRISPRi-based NOT gate library.**

A   Schematic of our engineered sgRNA:promoter pairs. Rainbows indicate specificity-determining regions with variable sequences.

B   Core sequences of P1–P9.

C   Orthogonality of $Si$:$Pi$ pairs. Each sgRNA was strongly induced, and the resulting fold change in sfGFP fluorescence from each promoter was measured.

D   Schematic of aTc-NOT$i$ circuits used to measure NOT gate transfer functions.

E   Transfer functions of NOT1–NOT9. Bacteria carrying aTc-NOT$i$ circuits were grown at different aTc concentrations. Input strains used sfGFP to report $IN_{NOTi}$ and output strains similarly reported $OUT_{NOTi}$. Mean sfGFP fluorescence was averaged across three replicates measured on different days. Error bars represent the standard error of the mean (s.e.m.) of mean sfGFP fluorescence. Dashed lines indicate maximum gate output in the absence of the sgRNA gene. Colored lines represent model fits. Fit parameters are shown in Appendix Table S1. Error (E) represents RMSE between fit and data in MEFL decades.

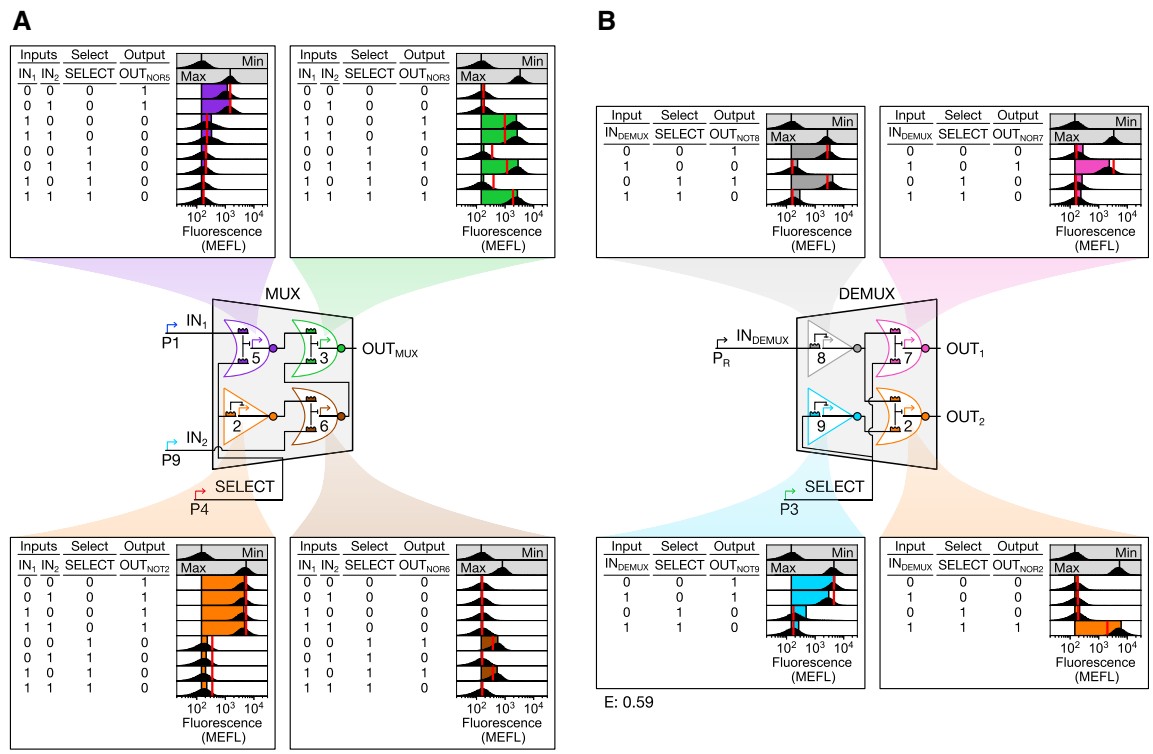

**Figure 3. Construction of biological MUX and DEMUX circuits.**

A  MUX design and characterization.
B  DEMUX design and characterization. The strong promoter $P_R$ was selected to generate $IN_{DEMUX}$ after P1 proved too weak to deactivate NOT8. P1 effectively generates $IN_1$ in the MUX, though, because NOR5 is more sensitive than NOT8 (Appendix Table S1).

Data information: Flow cytometry fluorescence distributions (black), model-simulated mean fluorescence (vertical red lines), and experimental mean sfGFP fluorescence (colored bars) are shown. Mean sfGFP fluorescence bars are bounded at left by mean autofluorescence and at right by mean fluorescence. RMSE between simulated and measured mean sfGFP fluorescence (E) is expressed in MEFL decades. Min and max indicate cellular autofluorescence and gate output in the absence of sgRNA, respectively (vertical lines indicate mean). Min was measured in triplicate on three separate days, max was measured once on a fourth day, and all other measurements were performed on a fifth day.

mean square error (RMSE) between the data and fits range between 0.067–0.30 MEFL decades (Materials and Methods).

Each NOT$i$ gate can be converted into a corresponding NOR gate (i.e., NOR$i$) by adding a second instance of the sgRNA transcribed from a second, independent input promoter (Appendix Fig S4A). NOR gates produce high output only when both inputs are low. We simulated the 2-input/1-output transfer functions of NOR$i$ by adding a second transcriptional input term to each NOT$i$ model (Appendix Text S3). Indeed, all nine NOR$i$ gates are predicted to exhibit NOR-like behavior (Appendix Fig S4B). We therefore hypothesized that our library of NOT and NOR gates could be used to construct the MUX and DEMUX.

### Construction and validation of MUX and DEMUX sub-circuits

We selected NOR5, NOR3, NOT2, and NOR6 to implement the MUX (Fig 3A). Considering only low (0) and high (1) signal values, a 2-input MUX can receive eight combinations of $IN_1$, $IN_2$, and SELECT. To generate these eight input combinations, we constructed eight MUX test circuits wherein constitutive promoters are absent (resulting in low signal) or present

(resulting in high signal) in the appropriate circuit locations (Fig 3A). Then, we probed the output of each gate and the overall MUX behavior in all eight test circuits. First, all gates propagate digital-like signals, generating only the minimum or maximum output value in all conditions (Fig 3A). Second, all gates perform the proper computation in all test circuits (Fig 3A). As a result, the MUX correctly chooses to relay the $IN_1$ signal (to $OUT_{MUX}$) when SELECT = 0 and $IN_2$ when SELECT = 1 (Fig 3A). Finally, all gate outputs are predicted accurately by a MUX model constructed from the individual gate models (RMSE between model predictions and data = 0.33 MEFL decades) (Fig 3A, Appendix Text S4).

We similarly constructed the DEMUX from NOT8, NOR7, NOT9, and NOR2 (Fig 3B). A 2-output DEMUX can receive four combinations of input ($IN_{DEMUX}$) and SELECT signal values. We tested our DEMUX design using four test circuits, as before. All gates behave digitally, perform the proper computations, and are well predicted by a model (RMSE = 0.59 MEFL decades) (Fig 3B, Appendix Text S5). Indeed, the DEMUX correctly relays $IN_{DEMUX}$ to $OUT_1$ when SELECT = 0, and to $OUT_2$ when SELECT = 1 (Fig 3B). Thus, our MUX and DEMUX both function as designed.

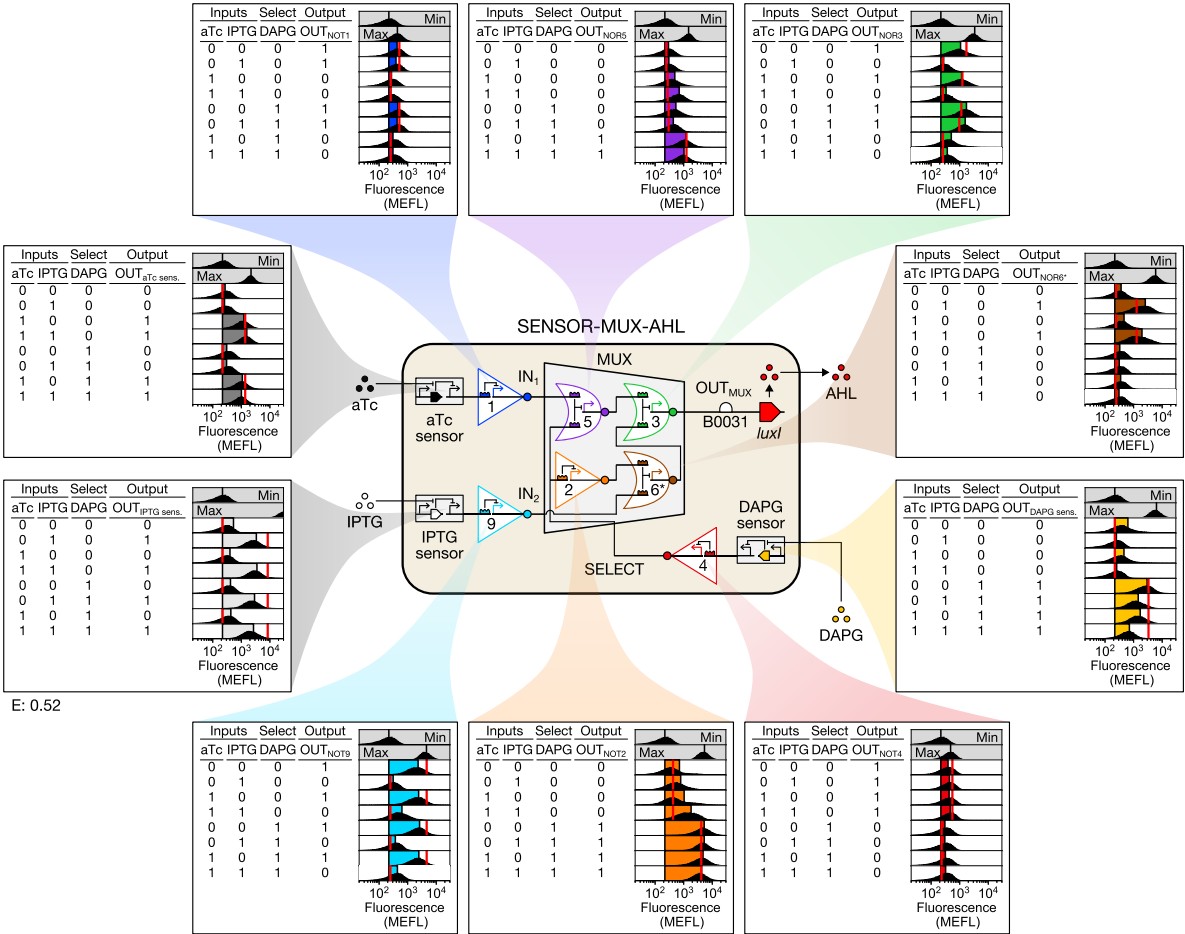

**Figure 4. Linking the MUX to chemical sensors and a cell-cell communication system.**

MUX $IN_1$, $IN_2$, and SELECT signals were connected to aTc, IPTG, and DAPG sensors, respectively, the MUX output was used to express the AHL biosynthetic enzyme LuxI, and NOR6 was replaced with the stronger NOR6* to correct an emergent fault, resulting in SENSOR-MUX-AHL. The output of every SENSOR-MUX-AHL sensor and gate was then characterized with the eight possible binary combinations of aTc, IPTG, and DAPG (Materials and Methods).

Data information: Data are visualized as in Fig 3. Min was measured once on one day, max was measured once on a second day, and all other measurements were performed on a third day. For sensors, max shows sensor output when the repressor is absent (aTc and DAPG sensors) or expressed from the genome only (IPTG sensor). Inducer concentrations: 0 (0); 20 ng/ml aTc, 0.3 mM IPTG, and 100 μM DAPG (1).

## Linking the MUX to small molecule inducers and the cell-cell communication system

A biological CS must link a MUX to a DEMUX using a cell-cell communication channel, and it should ideally interface with regulated promoters—those whose activities vary in response to perturbations. To this end, we first constructed SENSOR-MUX-AHL, wherein the aTc sensor controls $IN_1$, an isopropyl β-D-1-thiogalactopyranoside (IPTG) sensor controls $IN_2$, and a 2,4-diacetylphloroglucinol (DAPG) sensor controls SELECT (Fig 4, Appendix Fig S5). Each sensor controls its input via an additional sgRNA, thereby routing it through an additional NOT gate, so $IN_1$ = NOT(aTc), $IN_2$ = NOT(IPTG), and SELECT = NOT(DAPG). For cell-cell communication, we chose the widely used 3-oxohexanoyl acylhomoserine lactone (AHL) system, placing production of the AHL biosynthetic enzyme LuxI under the control of $OUT_{MUX}$ (Fig 4, Appendix Fig S6). While constructing SENSOR-MUX-AHL, we discovered that NOR6 produces

too low an output signal to fully repress NOR3 (the MUX output gate) in the context of the newly added sensors (Appendix Fig S7). We corrected this fault by increasing the strength of P6, yielding NOR6* and recovering proper NOR3 repression (Appendix Fig S8). Performance of SENSOR-MUX-AHL closely matches predictions from a model based on individual sensors and gates (RMSE = 0.52 MEFL decades) (Fig 4, Appendix Text S6).

## Linking the DEMUX to the cell-cell communication system and a small molecule inducer

We then constructed AHL-DEMUX, wherein an AHL sensor controls $IN_{DEMUX}$ and DAPG again controls SELECT (Fig 5). We found the DAPG sensor used in SENSOR-MUX-AHL too weak to effectively control the DEMUX SELECT signal (Appendix Fig S9) and instead used a slightly stronger DAPG sensor developed previously ("DAPG sensor 2;" Appendix Fig S10, Appendix Table S3). We also

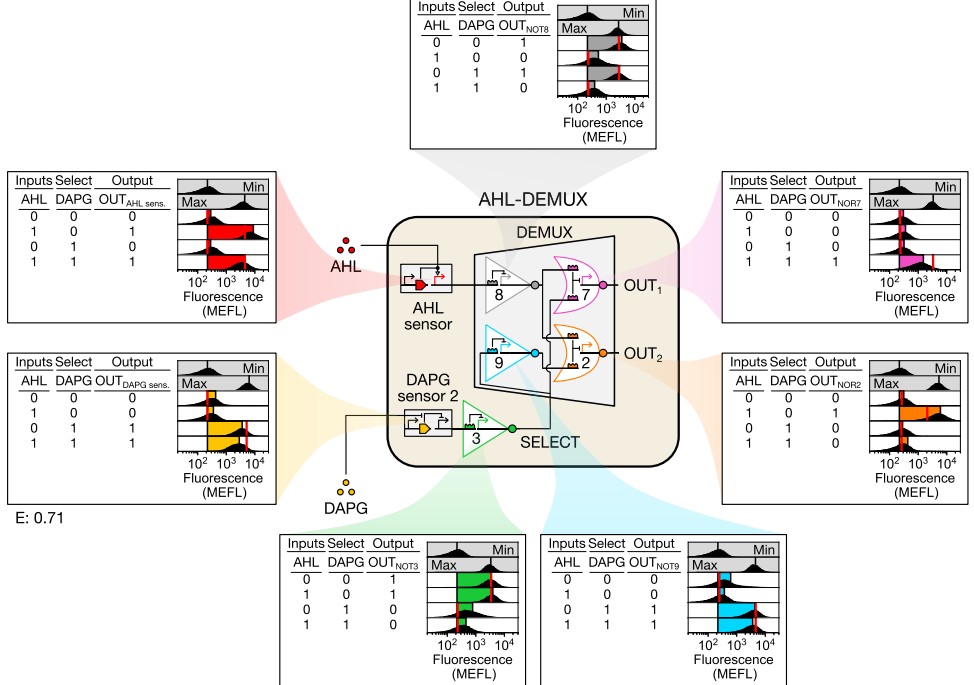

**Figure 5. Linking the DEMUX to the cell-cell communication system.**

The DEMUX input (IN_DEMUX) was connected to an AHL sensor, and SELECT was again connected to a DAPG sensor, as in SENSOR-MUX-AHL, resulting in AHL-DEMUX. A slightly stronger DAPG sensor ("DAPG sensor 2") was required to correctly control the DEMUX SELECT signal. Every AHL-DEMUX sensor and gate output was then characterized with the four possible binary combinations of AHL and DAPG (Materials and Methods).

Data information: Data are visualized as in Fig 3. Min was measured once on one day, max was measured once on a second day, and all other measurements were performed on a third day. For sensors, max shows sensor output when the repressor is absent (DAPG sensor) or at maximum induction (AHL sensor). Inducer concentrations: 0 (0); 100 nM AHL, 100 μM DAPG (1).

---

discovered that strong induction of the AHL sensor erroneously activated both DEMUX outputs, possibly due to saturation of dCas9 with the AHL-induced S8 sgRNA (Appendix Fig S11). To address this, we reduced the strength of the AHL sensor by mutating its output promoter (P_lux) (Appendix Fig S12), which recovered proper activation of only the correct DEMUX output in response to AHL (Fig 5, Appendix Fig S13). Probe experiments confirm that AHL-DEMUX computes proper outputs for all four possible AHL and DAPG input combinations, and the behavior of every sensor and gate agrees with model predictions (RMSE = 0.71 MEFL decades) (Fig 5, Appendix Text S7).

**Implementing the biological channel selector**

We implemented the full CS by growing SENSOR-MUX-AHL and AHL-DEMUX cells separately with the eight combinations of aTc, IPTG, and DAPG, and then mixing them into co-cultures and regularly diluting them with fresh media (Materials and Methods). In each case, we monitored the conversation by probing OUT_MUX, OUT_1, and OUT_2. Indeed, when DAPG is present, only conversation 1 occurs: The presence of aTc is relayed through both circuits and cell strains, ultimately controlling OUT_1 (Fig 6A). Furthermore, when DAPG is absent, conversation 2 occurs instead: IPTG is relayed through both circuits and strains, and controls OUT_2 (Fig 6B). These results indicate that SENSOR-MUX-AHL cells link correctly with AHL-DEMUX cells in co-culture.

To model the full CS, we developed gene expression dynamics models of SENSOR-MUX-AHL and AHL-DEMUX and linked them via an AHL production model. The gene expression dynamics models assume excess dCas9 and stable, quickly forming dCas9: sgRNA complexes (Appendix Text S8), and the AHL production model assumes stable LuxI, whose AHL production rate in SENSOR-MUX-AHL cells we measured using an AHL RECEIVER strain as a biosensor (0.089 nM/h/OD_600/MEFL; Appendix Fig S14, Materials and Methods, Appendix Text S9). The resulting CS simulation closely matches observed CS behavior (RMSE = 0.58 MEFL decades, Fig 6, Appendix Text S10).

**Characterizing CS signal propagation dynamics**

The longest computation path through the CS comprises eight sequential layers: the DAPG sensor, NOT4, NOT2, NOR6*, NOR3, the AHL system, NOT8, and NOR7 (Appendix Fig S15A). We characterized the dynamics of signal propagation through this path by adding DAPG to co-cultured SENSOR-MUX-AHL and AHL-DEMUX cells and probing the outputs of each layer over time (Appendix Fig S15B and C). The DAPG sensor responds immediately, followed in turn by SENSOR-MUX-AHL gate reporters, with the output gate (NOR3) activating between 5 and 7 h post-induction (Appendix Fig S15B). The AHL sensor activates simultaneously with NOR3, suggesting rapid communication, followed by AHL-DEMUX gate reporters, with the OUT_1 reporter activating ~11 h post-induction

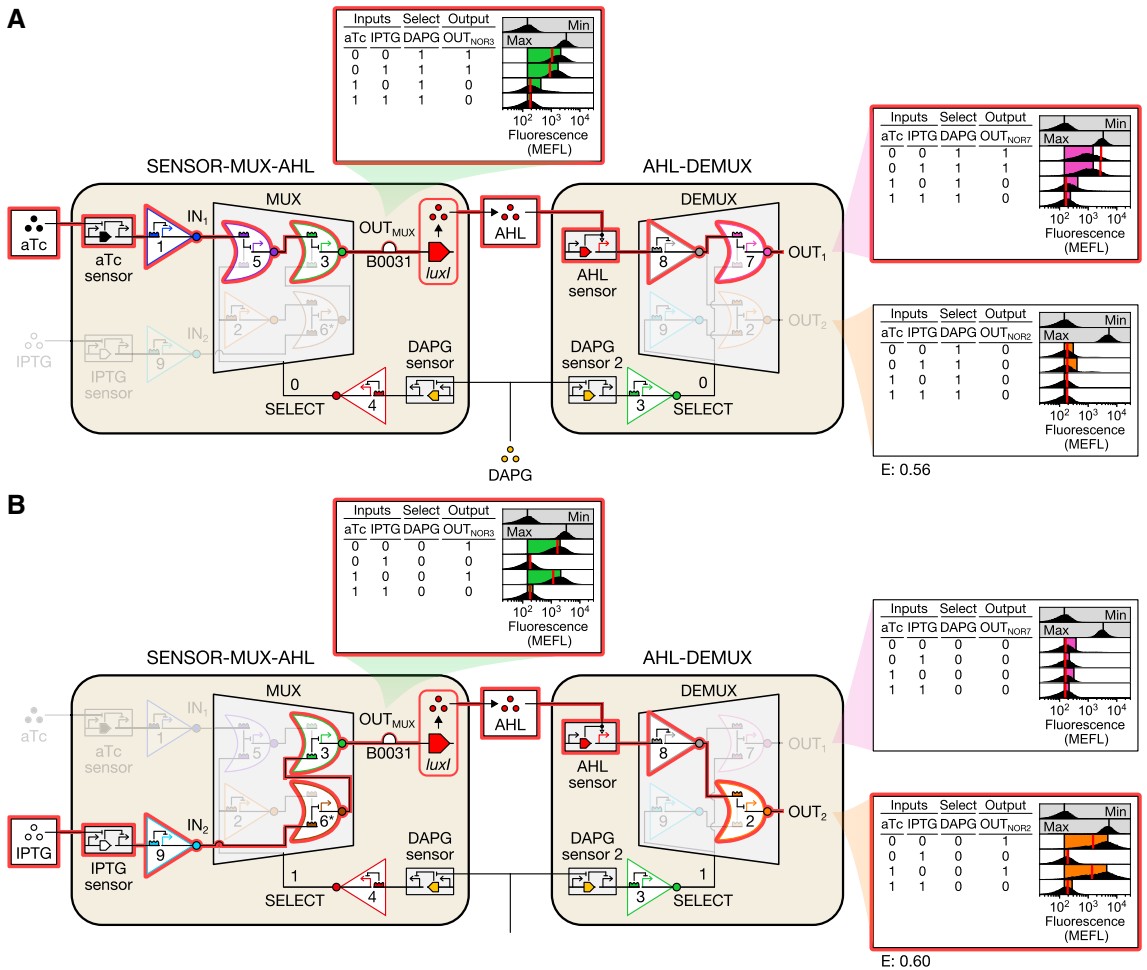

**Figure 6.  Combining SENSOR-MUX-AHL and AHL-DEMUX.**

A    Conversation 1. In the presence of DAPG, low aTc results in high $IN_1$, which is routed to $OUT_1$ via AHL.

B    Conversation 2. In the absence of DAPG, low IPTG results in high $IN_2$, which is routed to $OUT_2$ via AHL.

Data information: Conversations are highlighted in red. Data are visualized as in Fig 3. Min was measured in triplicate on three separate days, max was measured once on a fourth day, and all other measurements were performed on a fifth day.

(Appendix Fig S15C). A dynamical model again closely predicts these results (RMSE = 0.54 MEFL decades) (Appendix Fig S15B and C, Appendix Text S10).

**Multiplexing cell-cell communication**

Finally, we demonstrated that our CS can transmit multiple biological conversations sequentially by changing SELECT with DAPG (Fig 7). We first conditioned SENSOR-MUX-AHL and AHL-DEMUX cells to select and activate conversation 1, as before, and then diluted cells with different inducers to select and activate conversation 2. We initially observed $IN_1$ and $OUT_1$ activate while $IN_2$ and $OUT_2$ remained inactive, as expected, and then after changing inducers, we observed the opposite: $IN_2$ and $OUT_2$ activated and $IN_1$ and $OUT_1$ deactivated, as expected (Fig 7). Moreover, the $OUT_1$ and $OUT_2$ signals recapitulate prior co-culture results (Figs 7 and 6), and the $IN_1$ and $IN_2$ signals match prior SENSOR-MUX-AHL characterization (Figs 7 and 4). These results also closely agree with

dynamical model simulations (RMSE = 0.53 MEFL decades) (Appendix Text S10).

# Discussion

Advanced multicellular behaviors will likely require more than two cell-cell communication channels (Abelson *et al*, 2000). Genetically encoded CSs can enable such applications by reusing communication systems. As the number of transcriptional logic gates that can be implemented in a single cell continues to increase, CSs that can switch between three or more conversations can be envisioned (Appendix Fig S16, Appendix Text S11). Our CRISPRi-based CS requires multiple hours to perform its computation, largely due to the slow decay dynamics of dCas9. In the future, CS computation times could be substantially accelerated using de-stabilized (e.g., proteolysis-tagged) repressors. Eventually, CSs could transmit information on the seconds timescale using logic gates based upon post-

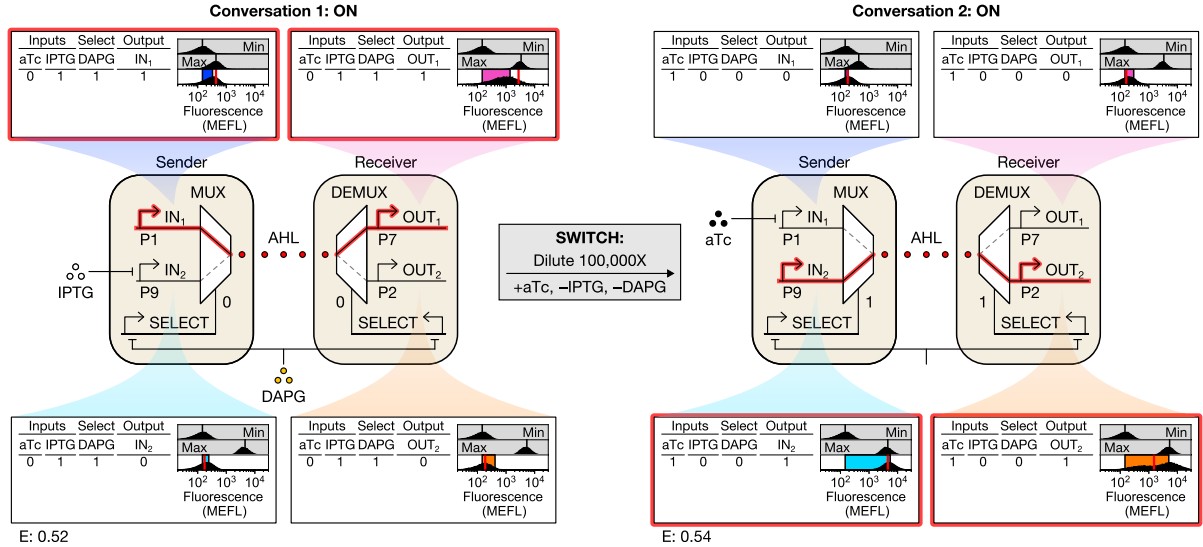

**Figure 7. Multiplexing cell–cell communication.**

First, conversation 1 occurs, wherein $IN_1$ activates and is transmitted over the AHL wire to $OUT_1$ while $IN_2$ and $OUT_2$ remain inactivate. Then, cells were diluted with different chemical inducers to switch the CS to conversation 2, wherein $IN_2$ activates and is transmitted over the same AHL wire to $OUT_2$ while $IN_1$ and $OUT_1$ deactivate.

Data information: Data are visualized as in Fig 3. Min was measured in triplicate on three separate days, max was measured once on a fourth day, and all other measurements were performed on a fifth day.

translational, rather than transcriptional regulation (Olson & Tabor, 2012; Gao *et al*, 2018; Chen *et al*, 2020). Such advances will enable major progress toward long-standing goals of synthetic biology such as programmed cellular differentiation, multicellular pattern formation, and the coordination of multiple metabolic pathway activities between a set of strains in a bioreactor.

# Materials and Methods

### Design of sgRNA:promoter pairs

We designed our sgRNAs to specifically repress their cognate promoters without cross-repressing essential *E. coli* MG1655 genes. To this end, we first generated random pre-sequences encoding the 3′-most 14 nucleotides of a CRISPRi operator, which includes the specificity-conveying seed region, followed by 1 nucleotide encoding the first (degenerate) nucleotide of the adjacent PAM site (NGG). We biased these pre-sequences for ~40% GC content. We then appended two guanine residues to the 3′-end of each pre-sequence to complete the PAM site. Then, we designed two different promoter sequences from each pre-sequence:GG by flanking it with −35 and −10 sites from apFAB126 (Mutalik *et al*, 2013) and λ $P_R$. We designed one sgRNA for each promoter by encoding six nucleotides matching the promoter −35 site, 14 nucleotides matching the 5′-end of each pre-sequence, and 82 nucleotides encoding the *Streptococcus pyogenes* Cas9 handle and terminator (Fig 2A). Promoters containing more than one additional primary and any additional secondary (NAG) PAM sites relative to the −35 and −10 sequences from which they were made were discarded, as were other promoters and sgRNAs derived from the same pre-sequence. Promoters

containing BsaI restriction sites, which interfere with Golden Gate cloning, were also discarded along with related promoters and sgRNAs.

We then computationally screened each sgRNA for off-target binding to the *E. coli* MG1655 genome (GenBank U00096.3). First, we used custom Python scripts to search for genome sequences overlapping essential genes and matching the 3′-most 10 or more nucleotides of the operator:PAM sequence from each promoter. Essential genes were identified by querying all MG1655 gene features against the EcoCyc *E. coli* database (https://ecocyc.org/). If cell growth was reported to be arrested when the gene was knocked out in any growth medium, the gene was considered essential. Genome features of type "rRNA," "tRNA," and "rep_origin" were also considered essential. If overlap of an essential feature was detected, the sgRNA:promoter pair was discarded, along with other pairs sharing a pre-sequence. Next, we searched for genome sequences matching the 3′-most 12 or more nucleotides of the operator:PAM sequence and again discarded any matches and pairs sharing a pre-sequence. Finally, we used the CasValue algorithm to identify sgRNAs likely to bind the genome (preprint: Aach *et al*, 2014). CasValue scores sgRNA:genome binding based on adjacency to PAM sites and sequence mismatches inside and outside the sgRNA seed region (defined as the 3′-most 13 nucleotides of the operator sequence). To use CasValue, we first compiled the *E. coli* genome into Bowtie index files via the bowtie-build command from Bowtie version 1.0.1 (Langmead *et al*, 2009). We then used default scoring and *S. pyogenes* Cas9 settings and specified that no sgRNA operators should be permitted in the genome ("accept" parameter = 0). Given these parameters, CasValue identified sgRNAs with seed sequences that matched the genome exactly or contained one mismatch and were adjacent to a primary or secondary PAM site.

Any identified sgRNAs were discarded along with other sgRNAs and promoters sharing a pre-sequence. Ultimately, we designed 3,000 sgRNA:promoter pairs derived from $P_R$ and 3,000 derived from apFAB126 meeting these criteria.

Next, we used a graph theoretic approach to select two subsets of 10 dissimilar sgRNA:promoter pairs (one set derived from $P_R$ and the other from apFAB126). To achieve this, we enriched the 3,000 pre-sequences encoded in the 6,000 designed promoters for mismatches. We represented sets of pre-sequences as complete graphs wherein vertices represented pre-sequences and edges were weighted by number of co-occurring nucleotides (i.e., nucleotide matches). Framed this way, we sought a 10-vertex subgraph that minimized the sum of all edge weights. The global optimum for this problem cannot be found quickly, so we developed several heuristics-based approaches to expedite our search. To begin, we sorted the ~4.5 million edges of our 3,000-vertex pre-sequence graph and then collected 100 vertices starting from edges with the lowest weights (fewest matches). Then, we iteratively reduced the total number of pairwise nucleotide matches (i.e., sum of all edge weights) among the 100 selected pre-sequences by looping four times over the 2,900 excluded pre-sequences, incorporating one pre-sequence at a time (resulting in a 101-vertex graph), and then discarding the pre-sequence contributing the most matches to the total. The 100-pre-sequence subset resulting from this procedure contained 20,096 total matches with 4.1 matches per edge on average and maximum 13 matches (of 15 possible). Next, we reduced this set to multiple 10-pre-sequence subsets by starting with all sets of two pre-sequences containing 5 or fewer sequence matches (i.e., edges with weight ≤ 5) and then randomly selecting a set to expand. Sets were expanded by separately incorporating each excluded pre-sequence and then retaining expanded sets that maintained 5 or fewer matches between all pairs of sequences. 8,154 sets of size 10 or greater were generated using this approach (with the largest set containing 14 pre-sequences). To select a final set, we developed a simple linear scoring algorithm that punished sequence matches at the 3′-end of the pre-sequence. We then selected the set with the lowest average score per edge. From this set, we created 10 sgRNA: promoter pairs based on $P_R$ and 10 based on apFAB126.

The strength of all twenty corresponding promoters was then characterized using sfGFP and flow cytometry. We discovered that most of the $P_R$-derived promoters were significantly weaker than $P_R$. We therefore selected the apFAB126-derived promoters for all but one pre-sequence (corresponding to P6), where the $P_R$-derived promoter was closer in strength to the other nine in the set than the apFAB126-derived alternative. We observed that one of the 10 sgRNAs slowed growth upon induction (guide sequence = TCGACACAACGTGACGTATG), so we discarded it to arrive at nine orthogonal sgRNA:promoter pairs.

### Promoter insulation

To reduce the influence of DNA context on promoter behavior (Davis *et al*, 2011; Nielsen *et al*, 2016), we fused randomly generated putatively inert insulator sequences upstream of core promoter sequences for P1–P9, $P_{tet}$, $P_{tac}$, and the constitutive promoters driving *phlF* (BBa_J23108, BBa_J23114) and *luxR* (BBa_J23115). Seventy-nucleotide insulator sequences were randomly generated and then screened against a list of forbidden *E. coli* sigma factor

binding sites, RBS features, transposon insertion sites, common restriction enzyme recognition sites, and unwieldy sequence repeats (Casini *et al*, 2014) using custom Python scripts. We also forbade the TetR O2 operator sequence, a sequence encoding the LacI operator site and a small portion of the LacI promoter, the native *S. pyogenes* Cas9 handle sequence, and the sequence of a structurally optimized handle (the "F+E" handle from (Chen *et al*, 2013a)). Insulators were then paired with P1-P9 using promoter prediction tools (Reese, 2001; Solovyev & Salamov, 2011) to ensure additional unwanted promoters were not introduced. The $P_{tet}$, $P_{tac}$, BBa_J23108/BBa_J23114, and BBa_J23115 insulators were then chosen randomly from the remaining available insulator sequences. If necessary, insulators were truncated to achieve a total promoter length of 105 base pairs.

### Strains, media, antibiotics, and chemical inducers

*Escherichia coli* NEB 10-β was used for cloning, and *E. coli* MG1655 was used for experiments. LB Miller medium (MilliporeSigma 1.10285.5000 or Fisher BioReagents BP9723-5) was used to maintain cloning strains, and M9 medium (M9 salts (6.78 g/l $Na_2HPO_4$, 3 g/l $KH_2PO_4$, 0.5 g/l NaCl, 1 g/l $NH_4Cl$ via BD 248510 or 5.962 g/l $Na_2HPO_4$, 3.266 g/l $KH_2PO_4$, 0.526 g/l NaCl, 1.016 g/l $NH_4Cl$ via Teknova M1902), 0.1 mM calcium chloride (Fisher Chemical C79-500), 2 mM magnesium sulfate (VWR BDH9246-500G), 0.4% glucose (Avantor Performance Materials 4908-06), and 0.2% casamino acids (MilliporeSigma 2240-500GM)) buffered to pH 6.6 with 100 mM HEPES (MilliporeSigma 5320-500GM) was used for all experiments except the orthogonality assay, where unbuffered M9 lacking HEPES was used. Plasmids with a ~15-copy ColE1, ~5-copy p15A, or ~3-copy pSC101* origin of replication (Segall-Shapiro *et al*, 2018) were used. ColE1 plasmids bear a chloramphenicol resistance marker and were maintained with 35 μg/ml chloramphenicol (Alfa Aesar B20841). p15A plasmids bear a spectinomycin antibiotic resistance marker and were maintained with 100 μg/ml spectinomycin (Gold Biotechnology S-140-25). pSC101* plasmids bear an ampicillin resistance marker and were maintained with 50 μg/ml ampicillin (Dot Scientific DS102040). aTc (Clontech 631310), IPTG (IBI Scientific IB02125), DAPG (Santa Cruz Biotechnology 2161-86-6), and AHL (Sigma-Aldrich, K3007-10MG) were used for chemical induction. aTc was dissolved in ethanol, IPTG was dissolved in filtered water and then filter-sterilized with a 0.2 μm syringe filter, DAPG was dissolved in 200 proof ethanol and then filter-sterilized with a 0.2 μm syringe filter, and AHL was dissolved in filtered water.

### Plasmid assembly

Plasmids were constructed using Golden Gate assembly (Engler *et al*, 2008). DNA fragments were first amplified using primers containing unique Golden Gate assembly junctions (Integrated DNA Technologies, Coralville, IA) and then size-separated via gel electrophoresis. Correctly sized bands were purified (Promega Wizard Gel Clean-Up kit, A9282) and then assembled into bacterial plasmids via digestion-ligation reactions with BsaI and T4 DNA ligase (New England Biolabs; BsaI = R3535 or R3733, T4 = M0202). All plasmid sequences were designed to lack BsaI sites. Hierarchical assembly, where parts were assembled into intermediate linear

fragments and then reamplified, was frequently used. Plasmid inserts were verified via Sanger sequencing (GeneWiz or Lone Star Labs).

*cas9* was amplified from pMJ806 (Jinek *et al*, 2012). *dcas9* was created by introducing the nuclease-deactivating mutations D10A and H840A (Qi *et al*, 2013) during plasmid construction. *dcas9* was expressed weakly from pSC31_1, which was used for the orthogonality assay, and strongly from pSC31_3, which was used for all other experiments (Appendix Fig S17). The aTc and IPTG sensors were sourced from pSR11-2 (Ramakrishnan & Tabor, 2016; Appendix Figs S5 and S18). The DAPG sensors are based on a previously published version (Nielsen & Voigt, 2014; Appendix Figs S5 and S18). Terminator sequences within the IPTG and DAPG sensors were replaced with strong synthetic versions (Chen *et al*, 2013b). The DAPG sensor was initially inactive and unresponsive to DAPG, so we replaced the strong BBa_J23101 promoter expressing *phlF* with the weak BBa_J23114 promoter, yielding the maximally activating DAPG-responsive "DAPG sensor 2." DAPG sensor 2 exhibited leaky expression in some conditions, however (Appendix Fig S7), so we replaced BBa_J23114 with the moderate-strength BBa_J23108 promoter, resulting in a non-leaky, near-maximally activating DAPG sensor (Appendix Figs S5 and S18, Appendix Table S3). *luxI* and *luxR* were amplified from pJT105 (Tabor *et al*, 2009). NOT and NOR gates were designed from sgRNA:promoter pairs and strong, sequence-distinct terminators (Chen *et al*, 2013b; Fig 2D and Appendix Fig S4A, terminators omitted for clarity). Circuit sequences were designed by fusing gate transcription units. Promoter prediction tools (Reese, 2001; Solovyev & Salamov, 2011) were used to identify and eliminate unwanted promoters from circuit sequences. 102-nucleotide putatively inert sgRNA placeholders were fused to DEMUX output promoters. Placeholders were created by randomly joining and truncating two unused 70-nucleotide promoter insulators. Gate transcription units were synthesized as gBlocks (Integrated DNA Technologies, Coralville, IA) and then incorporated into TOPO TA plasmids (Life Technologies, 450030) and sequence-verified prior to subsequent assembly. All plasmids used in this study are listed in Appendix Table S5.

### Freezer aliquots and experimental pre-cultures

All experimental cultures were started from $-80°C$ freezer aliquots to decrease day-to-day variability (Olson *et al*, 2017). To prepare aliquots, transformed strains were grown to exponential phase in LB, diluted 100-fold into buffered M9, and then grown for an additional ~4 h to $OD_{600} = ~0.1$. Cultures were then placed on ice, and glycerol was added to 18% final concentration. The culture-glycerol solution was then gently mixed, and 110 μl was aliquoted into PCR tubes, which were stored at $-80°C$. To initiate an experiment, a freezer aliquot was removed from $-80°C$ storage, thawed, and gently mixed before transferring a calculated volume ranging from ~1 to 110 μl into 3 ml fresh buffered M9. Inoculated pre-cultures were then grown shaking at 37°C to $OD_{600} = ~0.1$ and subsequently used to inoculate experimental cultures.

### Orthogonality assay

S1–S9 were expressed under control of the aTc sensor on separate ColE1 plasmids. P1–P9:*sfgfp* fusions were encoded on p15A plasmids. Pairwise combinations of these plasmids were co-transformed

with pSC31_1 to create 81 strains (Appendix Table S6). To begin the assay, 3 ml LB cultures containing appropriate antibiotics were inoculated from LB-glycerol freezer stocks. Cultures were then grown shaking at 37°C overnight. The following morning, LB cultures were diluted into 3 ml unbuffered M9 containing appropriate antibiotics and 0 or 20 ng/ml aTc to $OD_{600} = 5 \times 10^{-4}$. Induced M9 cultures were then grown shaking at 37°C for ~6 h to $OD_{600} = 0.3–0.6$. Cultures were then placed on ice, and 15 μl was transferred to 1 ml cold phosphate-buffered saline (PBS) and measured via flow cytometry.

### Probe plasmids

Probe plasmids were designed for P1–P9, P6*, $P_{tet}$, $P_{tac}$, $P_{PhlF}$, $P_{lux}$, $P_{lux*}$, and $P_R$. A standardized, insulated *sfgfp* gene was designed by fusing the self-cleaving ribozyme insulator RiboJ (Lou *et al*, 2012) to a synthetic RBS and the *sfgfp* coding sequence. This gene was then fused to promoters of interest and carried on plasmids with a p15A origin of replication (Appendix Fig S17). To probe a circuit promoter, we co-transformed the appropriate probe plasmid with a circuit plasmid and pSC31_3 (Appendix Fig S2). Entire circuits were probed by probing individual promoters in separate strains.

### NOT gate transfer functions

*Escherichia coli* were transformed with one of the nine aTc-NOT*i* plasmids, one input ($P_{tet}$) or output (P1–P9) probe plasmid, and pSC31_3. Pre-cultures were prepared from freezer aliquots and diluted to $OD_{600} = 8.84 \times 10^{-5}$ into fresh buffered M9 containing 0, 1, 1.5, 2, 3, 4, or 20 ng/ml aTc. These cultures were then grown shaking at 37°C for 5.75–6.5 h prior to fluorophore maturation and flow cytometry. An autofluorescence control strain lacking *sfgfp* and positive control strains lacking aTc-NOT*i* plasmids were similarly characterized at 0 ng/ml aTc.

### MUX and DEMUX characterization

Each MUX test circuit plasmid was separately co-transformed with P5, P3, P2, and P6 probe plasmids, resulting in 32 strains. Each DEMUX test circuit plasmid was co-transformed with P8, P7, P9, and P2 probe plasmids, resulting in 16 strains. All 48 strains also contained pSC31_3. Pre-cultures were prepared from freezer aliquots and then diluted into fresh buffered M9 to $OD_{600} = 8.84 \times 10^{-5}$ and grown shaking at 37°C for 5.75 h prior to fluorophore maturation and flow cytometry.

### Sensor transfer functions

*Escherichia coli* were transformed with one of five sensor circuit plasmids (Appendix Figs S5, S6D, S12B and S18), a probe plasmid containing $P_{tet}$, $P_{tac}$, $P_{PhlF}$, $P_{lux}$, or $P_{lux*}$, and pSC31_3. Pre-cultures were prepared from freezer aliquots and then diluted to $OD_{600} = 8.84 \times 10^{-5}$ into fresh buffered M9 containing inducer and grown shaking at 37°C for 5.75–6.5 h prior to fluorophore maturation and flow cytometry. aTc was induced to 0, 0.5, 0.793, 1.26, 1.99, 3.16, 5.01, 7.95, 12.6, and 20 ng/ml, IPTG was induced to 0, 0.0090, 0.0216, 0.0520, 0.125, 0.3, 0.721, 1.73, 4.16, and 10 mM, DAPG was induced to 0, 2.33, 4.36, 8.16, 15.3, 28.6, 53.5, 100, 187,

and 350 μM, and AHL was induced to 0, 0.1, 0.237, 0.562, 1.33, 3.16, 7.50, 17.8, 42.2, and 100 nM.

### Characterization of the AHL cell-cell communication system

Pre-cultures were prepared from freezer aliquots and then diluted into fresh buffered M9 to $OD_{600} = 0.001$ for SENDER cells and $OD_{600} = 8.84 \times 10^{-5}$ for RECEIVER cells. This difference in initial cell density (~3.5 doublings) accounts for the slower growth of SENDER cells due to *luxI* expression. Co-cultures crossing all four SENDER strains with all four RECEIVER strains were then grown shaking at 37°C for 5.75 h prior to fluorophore maturation and flow cytometry. We found cells well separated by their FL3 (mCherry) fluorescence and used a threshold of 1,000 MECY to classify SENDERs (above) and RECEIVERs (below).

### Characterization of SENSOR-MUX-AHL, AHL-DEMUX, and their variants

The SENSOR-MUX-AHL and SENSOR-MUX circuit plasmids were co-transformed with $P_{tet}$, $P_{tac}$, $P_{PhlF}$, P1, P2, P3, P4, P5, P6* or P6, and P9 probe plasmids, resulting in ten SENSOR-MUX-AHL and ten SENSOR-MUX characterization strains. The AHL-DEMUX, AHL-DEMUX v0.1, and AHL-DEMUX v0.2 circuit plasmids were similarly co-transformed with $P_{lux*}$ or $P_{lux}$, $P_{PhlF}$, P2, P3, P7, P8, and P9 probe plasmids, resulting in seven AHL-DEMUX, seven AHL-DEMUX v0.1, and seven AHL-DEMUX v0.2 characterization strains. All strains carried pSC31_3.

Pre-cultures were prepared from freezer aliquots and then diluted into fresh buffered M9 containing inducers to $OD_{600} = 5.52 \times 10^{-6}$ (SENSOR-MUX-AHL and SENSOR-MUX) or $OD_{600} = 8.84 \times 10^{-5}$ (AHL-DEMUX, AHL-DEMUX v0.1, and AHL-DEMUX v0.2). SENSOR-MUX-AHL and SENSOR-MUX cells were induced with all eight combinations of aTc (0 and 20 ng/ml), IPTG (0 and 0.3 mM), and DAPG (0 and 100 μM), and AHL-DEMUX, AHL-DEMUX v0.1, and AHL-DEMUX v0.2 cells were induced with all four combinations of AHL (0 and 100 nM for AHL-DEMUX, 0 and 3.5 nM for AHL-DEMUX v0.1 and v0.2) and DAPG (0 and 100 μM). AHL-DEMUX and AHL-DEMUX v0.2 cells were further characterized with 0, 1, 10, 100, and 1,000 nM AHL (Appendix Figs S11 and S13). Induced cultures were grown shaking at 37°C for 9 h (SENSOR-MUX-AHL and SENSOR-MUX) or 5.75–6.5 h (AHL-DEMUX, AHL-DEMUX v0.1, AHL-DEMUX v0.2) prior to fluorophore maturation and flow cytometry. SENSOR-MUX-AHL and SENSOR-MUX incubation times are longer to allow circuits to complete their computations. Prior experiments had also revealed that aTc was unable to induce cells for long periods of time, so we added a second dose (60 ng into 3 ml of culture) to +aTc SENSOR-MUX-AHL and SENSOR-MUX cultures 5.5–6 h into their 9 h incubation.

### Linking SENSOR-MUX-AHL and AHL-DEMUX cells in co-culture

We used the P3 probe plasmid to characterize SENSOR-MUX-AHL output ($OUT_{MUX}$), and the P7 and P2 probe plasmids to characterize the two AHL-DEMUX outputs ($OUT_1$ and $OUT_2$). All three strains carried pSC31_3. To begin, SENSOR-MUX-AHL and AHL-DEMUX pre-cultures were prepared from freezer aliquots and then induced

with all eight combinations of aTc (0 and 20 ng/ml), IPTG (0 and 0.3 mM), and DAPG (0 and 100 μM). +IPTG SENSOR-MUX-AHL cultures were diluted to $OD_{600} = 2.5 \times 10^{-6}$, −IPTG SENSOR-MUX-AHL cultures were diluted to $OD_{600} = 5 \times 10^{-6}$, and AHL-DEMUX cultures were diluted to $OD_{600} = 1.25 \times 10^{-6}$. These dilutions account for the increased duration of this experiment as well as minor differences in growth rate observed for different strains in different conditions. Induced cultures were then grown shaking at 37°C overnight for 12 h 20 min. until SENSOR-MUX-AHL cells reached ~0.25 $OD_{600}$ and AHL-DEMUX cells reached ~0.15 $OD_{600}$. As before, a second dose of aTc (60 ng into 3 ml of culture) was added to all +aTc cultures ~8 h into the overnight incubation.

Following overnight growth, preconditioned SENSOR-MUX-AHL and AHL-DEMUX cells were mixed together in fresh buffered M9 containing inducers and then diluted regularly. To inoculate each co-culture, SENSOR-MUX-AHL cells were diluted to $OD_{600} = 0.06162$ and AHL-DEMUX cells were diluted to $OD_{600} = 0.006162$. These dilutions were designed to achieve a 10:1 ratio of SENSOR-MUX-AHL cells to AHL-DEMUX cells and to maintain dense (~0.1–0.3 $OD_{600}$) co-cultures upon dilution every 1.5 h. SENSOR-MUX-AHL cells were prepared in duplicate, one co-culture for each AHL-DEMUX output strain. All 16 co-cultures were then grown shaking at 37°C and diluted with fresh media and inducers (840 μl co-culture into 2.16 ml warm media and inducers) every 1.5 h for 7.5 h (four dilutions total) prior to fluorophore maturation and flow cytometry. We found cells well separated by their FL3 (mCherry) fluorescence and used a threshold of 600 MECY to classify SENSOR-MUX-AHL (below) and AHL-DEMUX (above) cells.

### Measuring the AHL production rate of LuxI in SENSOR-MUX-AHL cells

$RECEIVER_{J23115*}$ cells were used as biosensors to quantify AHL produced by SENSOR-MUX-AHL cells over 2.5 h in the absence of inducers (wherein LuxI is strongly expressed from $OUT_{MUX}$). The P3 probe plasmid was used to quantify LuxI expression in SENSOR-MUX-AHL cells. First, a SENSOR-MUX-AHL pre-culture was prepared from a freezer aliquot and grown shaking at 37°C for ~10.5 h overnight. The next morning, a $RECEIVER_{J23115*}$ pre-culture was prepared from a freezer aliquot, and the SENSOR-MUX-AHL pre-culture was expanded by diluting it to $OD_{600} = 0.060$ in 10 ml buffered M9. After ~2 h of additional growth to ~0.3 $OD_{600}$, SENSOR-MUX-AHL cells were washed of AHL by pelleting them with a centrifuge (3,220 rcf, 4 min), discarding their supernatant, and resuspending them in 10 ml fresh media. A 40 ml culture was then inoculated to $OD_{600} = 0.0129$ and grown shaking at 37°C while 5 ml was sampled every 30 min. for the next 2.5 h (five samples total). From each sample, 30 μl was subjected to fluorophore maturation and flow cytometry to quantify SENSOR-MUX-AHL output, 125 μl was used to measure $OD_{600}$, and the rest of the sample was filtered through a 0.2 μm syringe filter to remove SENSOR-MUX-AHL cells and halt AHL production. 3 ml of filtered sample was then inoculated with $RECEIVER_{J23115*}$ cells to $OD_{600} = 8.84 \times 10^{-5}$ and grown shaking at 37°C for 6 h 10 min. prior to fluorophore maturation and flow cytometry. The $RECEIVER_{J23115*}$ pre-culture was diluted as necessary to maintain exponential growth while SENSOR-MUX-AHL samples were collected.

## Dynamical CS response to DAPG induction

$P_{PhlF}$, P4, P2, P6*, and P3 probe plasmids were used to characterize SENSOR-MUX-AHL, and $P_{lux*}$, P8, and P7 probe plasmids were used to characterize AHL-DEMUX. All eight strains carried pSC31_3. To begin, pre-cultures were prepared from freezer aliquots and then diluted to $OD_{600} = 2.5 \times 10^{-6}$ in fresh buffered M9 containing 0.3 mM IPTG (+IPTG). Cultures were then grown shaking at 37°C overnight for 12 h 20 min. until SENSOR-MUX-AHL cells reached ~0.5 $OD_{600}$ and AHL-DEMUX cells reached ~0.07 $OD_{600}$. Following overnight growth, preconditioned SENSOR-MUX-AHL and AHL-DEMUX cells were mixed together in fresh buffered M9 +IPTG to $OD_{600} = 0.09425$ (SENSOR-MUX-AHL) and $OD_{600} = 0.009425$ (AHL-DEMUX). These dilutions were designed to achieve a 10:1 ratio of SENSOR-MUX-AHL cells to AHL-DEMUX cells and to maintain dense (~0.1–0.3 $OD_{600}$) co-cultures upon dilution every 1 h. Seven co-cultures were prepared by pairing SENSOR-MUX-AHL and AHL-DEMUX strains probing different sensors and gates (strains probing the following promoters were paired: $P_{PhlF}$ and $P_{lux*}$, P4 and $P_{lux*}$, P2 and $P_{lux*}$, P6* and $P_{lux*}$, P3 and $P_{lux*}$, P3 and P8, P3 and P7). All seven co-cultures were then grown shaking at 37°C for 1 h to allow cells to acclimate. Then, co-cultures were diluted with fresh buffered M9 +IPTG containing enough DAPG to induce cultures to 100 μM DAPG (1.284 ml co-culture into 1.716 ml warm media and inducers). Co-cultures were then maintained shaking at 37°C for 12 h by diluting them every hour with fresh buffered M9 +IPTG +DAPG (1.284 ml co-culture into 1.716 ml warm media and inducers). Upon passage of each co-culture, 30 μl was collected and subjected to fluorophore maturation and flow cytometry. As before, a 600-MECY FL3 threshold was used to classify SENSOR-MUX-AHL (below) and AHL-DEMUX (above) cells.

## Multiplexing cell–cell communication

We used the P1 and P7 probe plasmids to characterize conversation 1 (probing $IN_1$ in SENSOR-MUX-AHL cells and $OUT_1$ in AHL-DEMUX cells, respectively), and the P9 and P2 probe plasmids to characterize conversation 2 (probing $IN_2$ in SENSOR-MUX-AHL cells and $OUT_2$ in AHL-DEMUX cells, respectively). All four strains carried pSC31_3. To begin, SENSOR-MUX-AHL and AHL-DEMUX cells were prepared from freezer aliquots and then diluted to $OD_{600} = 2.5 \times 10^{-6}$ (SENSOR-MUX-AHL) and $OD_{600} = 1.25 \times 10^{-6}$ (AHL-DEMUX) and induced with 0.3 mM IPTG (+IPTG) and 100 μM DAPG (+DAPG). Induced cultures were then grown shaking at 37°C overnight for 12 h and 20 min. until SENSOR-MUX-AHL cells reached ~0.3 $OD_{600}$ and AHL-DEMUX cells reached ~0.1 $OD_{600}$.

Following overnight growth, preconditioned SENSOR-MUX-AHL and AHL-DEMUX cells were mixed together in fresh buffered M9 +IPTG +DAPG and then diluted regularly with fresh media. SENSOR-MUX-AHL cells were inoculated to $OD_{600} = 0.066452$ and AHL-DEMUX cells were inoculated to $OD_{600} = 0.001329$ to achieve a 50:1 ratio of SENSOR-MUX-AHL cells to AHL-DEMUX cells and to maintain dense (~0.1–0.3 $OD_{600}$) co-cultures upon dilution. Two co-cultures were prepared, one monitoring conversation 1 (via strains probing $IN_1$ and $OUT_1$) and the other monitoring conversation 2 (via strains probing $IN_2$ and $OUT_2$). Both co-cultures were grown shaking at 37°C and diluted with fresh buffered M9 +IPTG +DAPG

(840 μl co-culture into 2.16 ml warm media) every 1.5 h for 7.5 h. Samples were then collected and subjected to fluorophore maturation and flow cytometry while co-cultures were diluted 100,000-fold in buffered M9 lacking IPTG and DAPG and induced with 20 ng/ml aTc (+aTc). Both co-cultures were then grown shaking at 37°C overnight for 13 h and 45 min. to ~0.5 $OD_{600}$. As before, a second dose of aTc (60 ng into 3 ml of culture) was added ~8 h into the overnight incubation. Both cultures were then diluted with fresh buffered M9 +aTc (840 μl co-culture into 2.16 ml warm media) every 1.5 h for 7.5 h (four dilutions total) prior to fluorophore maturation and flow cytometry. As before, a 600-MECY FL3 threshold was used to classify SENSOR-MUX-AHL (below) and AHL-DEMUX (above) cells.

## Fluorophore maturation

To allow nascent fluorescent proteins to mature and contribute to cellular fluorescence, 30 or 50 μl of liquid culture was added to 500 μl phosphate-buffered saline (PBS) containing 1 mg/ml chloramphenicol, which inhibits protein synthesis in bacteria, and then incubated in a 37°C water bath for 1 h (Olson *et al*, 2014, 2017). Samples were then placed on ice until flow cytometry was performed.

## Flow cytometry

Cellular fluorescence was measured using a modified BD FACScan flow cytometer. Samples were collinearly excited by blue (488 nm, 30 mW) and yellow (561 nm, 50 mW) solid-state lasers (Cytek), and sample fluorescence was acquired through an optical bandpass filter centered at 510 nm with a bandwidth of 21 nm (FL1) and separately through a 650 nm longpass filter (FL3). Forward scatter (FSC) and side scatter (SSC) fluorescence was also measured. Samples were acquired using either FlowJo Collectors' Edition 7.5.110.7 on a Windows 7 computer or BD CellQuest Pro 5.1.1 on a Macintosh computer. For *E. coli* samples, the voltage settings of the FSC, SSC, FL1, and FL3 detectors were FSC = 10X, SSC = 600, FL1 = 650, 750, or 850, and FL3 = 650, 750, or 850. FL1 and FL3 settings were adjusted to ensure cellular fluorescence did not saturate the detectors. An SSC threshold was specified in the acquisition software to eliminate low-SSC debris (below either 60% or 65% SSC). The fluorescence values of all events overcoming this threshold were recorded. Additionally, a polygon gate was manually drawn using the acquisition software to ensure enough *E. coli*-sized events were recorded. For monoculture experiments and characterization of the AHL cell–cell communication channel, 20,000–30,000 *E. coli*-sized events were recorded. For CS co-culture experiments, events were recorded for 1 min.

SpheroTech rainbow calibration particles (RCP-30-5A) were used together with the FlowCal Python package (Castillo-Hair *et al*, 2016) to calibrate FL1 and FL3 fluorescence measurements from arbitrary units (a.u.) to molecules of equivalent fluorescein (MEFL) and molecules of equivalent Cy5 (MECY), respectively. Calibrated fluorescence units enable comparison of measurements across instrument detector voltage and gain settings and account for long-term instrument drift (Castillo-Hair *et al*, 2016). For calibration particle samples, the FSC and SSC detector voltage settings were 10X and 460, respectively. The SSC threshold setting was 10%. FL1 and FL3

detector voltage settings were specified to match the settings used to measure *E. coli* samples during the same flow cytometry session. One or two drops of calibration particles were aliquoted from the manufacturer-issued dropper bottle into 500 µl or 1 ml phosphate-buffered saline (PBS) and then gently mixed and stored on ice until acquisition. Between 20,000 and 30,000 calibration particles were always acquired immediately prior to acquiring *E. coli* samples. After acquisition, FlowCal was used within custom Python scripts to generate MEFL and MECY calibration curves from calibration particle data. Calibration curves were then used to calibrate each FL1 and FL3 fluorescence measurement for all *E. coli* samples. FlowCal was also used to isolate *E. coli* cells from other particles. Specifically, a density gate was applied to all samples to retain events in the densest region of their FSC vs. SSC plots. The fraction of events retained was 0.85. Calibration particle samples were additionally gated to discard the first 250 and last 100 events and any events saturating the FSC or SSC channels, per default FlowCal behavior. Data from the orthogonality assay, which were taken prior to a major cytometer reconfiguration, were also gated to remove saturating events occurring in the lowest bin of the FL1 channel.

### Visualization of cytometry data

We visualized population-level cellular fluorescence using frequency distributions and violin plots. To create a frequency distribution, the fluorescence measurements of a sample, which initially span 1,024 possible MEFL values, are first down-sampled into 64 bins logarithmically spanning the same range of values. This assigns 1 bin per 16 adjacent MEFL values. The frequency of each bin is then plotted and normalized such that the most frequent bin (i.e., the peak) spans half the *y*-axis of a frequency distribution plot. To create a violin plot, cellular fluorescence is first binned into 125 bins logarithmically spaced from 10 MEFL to 1,000,000 MEFL (25 bins per decade). The frequency of each bin is then displayed using a normalized symmetrical horizontal bar plot ("violin"). The width of the violin indicates the relative frequency of different *y*-axis values. The top and bottom 1% of measurements were trimmed (discarded) for aesthetic purposes. The center vertical axis of each violin indicates its associated *x*-axis value. Mean cellular fluorescence, calculated prior to trimming, is also displayed atop each violin as a solid horizontal line.

### Error calculations

To compare circuit behavior to model fits or simulations, we calculated the root mean square error (RMSE) between predicted and measured $\log_{10}$-transformed sfGFP fluorescence values:

$$\text{RMSE} = \sqrt{\frac{\sum\limits_{i=1}^{N}\left[\left(\log_{10}(F_i^{\text{pred.}}) - \log_{10}(F_i^{\text{meas.}})\right)^2\right]}{N}}$$

where $F$ is sfGFP fluorescence and $N$ is the total number of measurements. RMSE summarizes the difference between predicted and measured sfGFP fluorescence in $\log_{10}$ space in units of MEFL decades. If a sfGFP fluorescence value was equal to zero (e.g., for some predicted sensor outputs), that comparison was omitted from the RMSE calculation.

### CS scaling laws

Scaling laws were determined that relate number of channels transmitted by a CS to number of CS gates, layers, maximum gate fan-in, and maximum gate fan-out (Appendix Fig S16). For these laws, the CS was assumed to be constructed from NOT and NOR gates, and its features were calculated assuming the MUX and DEMUX composing it were implemented using the same approach (i.e., 2-layer or Recursive). Laws were deduced by comparing different 2-, 4-, and 8-channel CS implementations and are listed in Appendix Text S11.

## Data availability

Measured and simulated mean cellular and sfGFP fluorescence values for the major experiments of this study are listed in Dataset EV1. Total sgRNA calculations from Appendix Figs S11 and S13 are listed in Table EV1. Python scripts performing the co-culture simulations described in Appendix Text S10 are included in Code EV1. Plasmid and DNA part sequences are included as GenBank files in Dataset EV2 and online via the Benchling platform (http://www.benchling.com).

**Expanded View** for this article is available online.

### Acknowledgements

We thank Evan Olson and Sebastián Castillo-Hair for helpful discussions on genetic circuit design, Dr. Joel Moake for use of his flow cytometer, Dr. Matthew Bennett for gifting us a plasmid harboring *cas9*, and Sebastián Castillo-Hair for constructing the dCas9-expressing pSC31_1 and pSC31_3 plasmids. This work was supported by the Office of Naval Research (YIP N00014-14-1-0487) and the U.S. National Science Foundation (NSF) (CAREER 1553317). JTS was also supported by the NSF Graduate Research Fellowship Program (GRFP) (DGE-0940902).

### Author contributions

JTS and JJT conceived of the study and designed the experiments. JTS performed the experiments and analyzed the data. JTS and JJT wrote the manuscript.

### Conflict of interest

The authors declare that they have no conflict of interest.

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
