## [Review Process File · Molecular Systems Biology]

Multiplexing cell-cell communication

John Sexton and Jeffrey Tabor
DOI: 10.15252/msb.20209618

Corresponding author(s): Jeffrey Tabor (jeff.tabor@rice.edu)

Review Timeline:

Submission Date:	7th Apr 20
Editorial Decision:	25th May 20
Revision Received:	2nd Jun 20
Accepted:	16th Jun 20

Editor: Maria Polychronidou

Transaction Report:

The reviewers' comments and authors' responses are not available with this article, as the initial review process took place with another journal.

25th May 2020

Manuscript Number: MSB-20-9618, Multiplexing cell-cell communication

Thank you again for submitting your work to Molecular Systems Biology. We have now heard back from the reviewer who was asked to evaluate your study. As we previously discussed, the reviewer was given access to the manuscript and your responses to the reviewers' comments from the other journal. They were asked to evaluate whether the reviewers' concerns have been adequately addressed, and to assess the suitability of the study for publication keeping in mind the editorial criteria of Molecular Systems Biology. As you will see below, the reviewer thinks that you have done an excellent job in addressing the previous concerns and is supportive of publication, pending some very minor modifications.

REFEREE REPORTS

Reviewer #1:

Sexton and Tabor engineered a channel selector synthetic gene circuit that switches between two intercellular conversations. The authors exhaustively tested its functionality in *E. coli* and demonstrated that the circuit is actuated as desired. The circuit is comprised of 12 CRISPRi-based transcriptional logic gates, 3 inducible promoters, and acyl homoserine lactone (AHL)-mediated quorum sensing system.

Borrowing its concept from electrical engineering, the presented genetically encoded channel selector links a multiplexer (MUX) and demultiplexer (DEMUX) circuit. MUX and DEMUX are constructed using several orthogonal NOT and NOR logic gates. They are implemented in separate bacterial strains, characterized individually, and co-cultured to validate their full functionality. The authors also demonstrate that their circuit is capable of switching between the two conversation dynamically.

All of the circuit behavior aligns with the authors' mathematical predictions, which further support the validity of the circuit function. The authors have done a very thorough job in responding to the comments/suggestions by the initial reviewers. Overall, I'm impressed with the work and support its publication.

Minor comments

1. The complexity, integrity, and performance of the circuit is fascinating. The authors did a stunning amount of work thoroughly characterizing and validating various components.
2. Because of the nature of the work, it can be confusing to follow through several negations in signal concentrations. It will help clarify if the authors could clearly state that inducer concentrations are not the inputs (IN1 or IN2) but instead the NOT(inducer) is the input to the channel selector.

We are happy to hear that the reviewer is positive about our manuscript and the edits we have made in response to the previous reviewers, and supports publication in *Molecular Systems Biology*.

In response to the new reviewer's second comment, we have changed the beginning of the "Linking the MUX to small-molecule inducers and the cell-cell communication system" section from:

"To this end, we first constructed SENSOR-MUX-AHL, wherein the aTc sensor controls IN1, an isopropyl β -D-1-thiogalactopyranoside (IPTG) sensor controls IN2, and a 2,4-diacetylphloroglucinol (DAPG) sensor controls SELECT, each via an additional sgRNA (Fig. 4, Appendix Fig. S5). For cell-cell communication, ..."

to

"To this end, we first constructed SENSOR-MUX-AHL, wherein the aTc sensor controls IN1, an isopropyl β -D-1-thiogalactopyranoside (IPTG) sensor controls IN2, and a 2,4-diacetylphloroglucinol (DAPG) sensor controls SELECT (Fig. 4, Appendix Fig. S5). Each sensor controls its input via an additional sgRNA, thereby routing it through an additional NOT gate, so $IN1=NOT(aTc)$, $IN2=NOT(IPTG)$, and $SELECT=NOT(DAPG)$. For cell-cell communication, ..."

16th Jun 2020

Manuscript number: MSB-20-9618R, Multiplexing cell-cell communication

Thank you for performing the requested text changes. We are now satisfied with the modifications made and I am pleased to inform you that your paper has been accepted for publication.

Corresponding Author Name: Jeffrey J. Tabor

Manuscript Number: MSB-20-9618